# The Emergence of Chikungunya ECSA Lineage in a Mayaro Endemic Region on the Southern Border of the Amazon Forest

**DOI:** 10.3390/tropicalmed5020105

**Published:** 2020-06-26

**Authors:** Carla Julia da Silva Pessoa Vieira, David José Ferreira da Silva, Janaína Rigotti Kubiszeski, Laís Ceschini Machado, Lindomar José Pena, Roberta Vieira de Morais Bronzoni, Gabriel da Luz Wallau

**Affiliations:** 1Health Sciences Institute, Federal University of Mato Grosso, Sinop 78550-728, MT, Brazil; cjspvieira@gmail.com (C.J.d.S.P.V.); silvadjf@gmail.com (D.J.F.d.S.); janainarigottik@gmail.com (J.R.K.); robertabronzoni@gmail.com (R.V.d.M.B.); 2Aggeu Magalhães Institute, Oswaldo Cruz Foundation, Recife 50670-420, PE, Brazil; laisceschini@gmail.com (L.C.M.); lindomar.pena@cpqam.fiocruz.br (L.J.P.)

**Keywords:** alphaviruses emergence, molecular epidemiology, human infection, arbovirus

## Abstract

Anthropic changes on the edges of the tropical forests may facilitate the emergence of new viruses from the sylvatic environment and the simultaneous circulation of sylvatic and urban viruses in the human population. In this study, we investigated the presence of arboviruses (arthropod-borne viruses) in the sera of 354 patients, sampled from February 2014 to October 2018 in Sinop city. We sequenced the complete genomes of one chikungunya virus (CHIKV)-positive and one out of the 33 Mayaro virus (MAYV)-positive samples. The CHIKV genome obtained here belongs to the East/Central/South African (ECSA) genotype and the MAYV genome belongs to the L genotype. These genomes clustered with other viral strains from different Brazilian states, but the CHIKV strain circulating in Sinop did not cluster with other genomes from the Mato Grosso state, suggesting that at least two independent introductions of this virus occurred in Mato Grosso. Interestingly, the arrival of CHIKV in Sinop seems to not have caused a surge in human cases in the following years, as observed in the rest of the state, suggesting that cross immunity from MAYV infection might be protecting the population from CHIKV infection. These findings reinforce the need for continued genomic surveillance in order to evaluate how simultaneously circulating alphaviruses infecting the human population will unfold.

## 1. Introduction

Arboviruses, arthropod-transmitted viruses, can successfully replicate in both invertebrates, vectors and vertebrate host cells [1]. These viruses are mostly transmitted by mosquitoes in sylvatic and urban environments, causing a large public health burden in several tropical countries around the globe [2]. WHO estimates that the dengue virus alone infects around 100–400 million people annually, mainly in the South American and Asian continents [3]. On the other hand, several human pathogenic arboviruses circulate in the sylvatic environment, with eventual spillover to the human population, such as Saint Louis encephalitis virus (SLEV) [4,5], West Nile virus (WNV) [6,7] and Mayaro virus (MAYV) [8,9].

Most known human-infecting arboviruses belong to the families Flaviviridae, Togaviridae and Peribunyaviridae [9,10]. Viruses from the Togaviridae family are zoonotic and epizootic pathogens that cause recurrent epidemics in both human and animal populations [11,12], such as the mosquito-borne alphaviruses chikungunya virus (CHIKV), O’nyong’nyong virus (ONNV), Ross River virus (RRV) and Mayaro virus (MAYV), and the viruses of the Western (WEEV), Eastern (EEEV) and Venezuelan (VEEV) equine encephalitides [12,13]. The arthritogenic alphaviruses MAYV and CHIKV were first isolated in the 1950s [14,15]. Since then, MAYV has been considered endemic and enzootic in South America and the Caribbean [16,17], while CHIKV emerged in Africa but spread globally, showing a much more complex evolutionary history. Phylogenomic analyses of CHIKV have revealed three distinct genotypes: West African (WA), East/Central/South African (ECSA) and Asian/Caribbean [18]. In Africa, both the WA and ECSA genotypes were circulating mainly enzootically among non-human primates, with occasional spillover to humans, while some ECSA lineages, such as the Indian Ocean lineage (IOL), expanded their range and have now been causing large outbreaks in the Asian continent. The Asian/Caribbean genotype is predominantly endemic/epidemic among human populations [19,20].

In Brazil, two CHIKV genotypes emerged in 2014: The Asian/Caribbean genotype in the Amapá state [21] and the ECSA genotype in the Bahia state [22]. Thereafter, a surge of human cases occurred; more than 500,000 cases have been reported by the Brazilian Ministry of Health [23]. Conversely, the neglected tropical Mayaro fever has no reports in the Brazilian epidemiological bulletins. However, studies have shown that 495 MAYV-infection cases have been detected in Brazil so far—more than half of all 901 cases reported in Latin America and the Caribbean [17]. These data show that the two viruses are constantly infecting the human population, with dramatically different dynamics but, until now, no studies have identified their co-circulation.

Chikungunya virus and MAYV circulate between an invertebrate vector and a vertebrate host [13], involving non-human primates and arboreal *Aedes* (for CHIKV) and *Haemagogus janthinomys* (for MAYV) mosquitoes [24,25]. Both the Asian/Caribbean and the ECSA lineages of CHIKV can circulate in urban cycles between humans and their main vectors *Aedes aegypti* and *Ae. albopictus*. Contrastingly, the WA lineage has never switched to an urban cycle and is maintained in a sylvatic cycle, including non-human primates and sylvatic *Aedes* mosquitoes, which are not anthropophilic but have caused sporadic human cases by biting humans in rural areas [13]. Similarly, MAYV causes disease after spillover transmission to nearby human populations inferred as incidental or dead-end hosts [26]. However, new evidence suggests that a sustained urban transmission is also possible since *Ae. aegypti* has been shown to transmit MAYV in laboratory conditions [27,28] and the MAYV genome has been detected in naturally infected *Ae. aegypti* specimens [29]. The high prevalence of *Ae. aegypti* in the urban environment in Brazil and other South American countries where Mayaro fever is endemic suggests that an urban cycle could be established [25]. Based on the vector species’ habitat suitability and current knowledge about these viruses’ distribution, it is possible that both alphaviruses spread and co-circulate, causing human outbreaks [30].

The whole-genome sequencing of clinical isolates during epidemics and interepidemic periods can be used to uncover the pathogen’s natural history and spread pattern, providing background information to guide monitoring and prevention strategies, foster vaccine development and improve diagnostics and treatment [31]. The study of full-length arboviral genomic sequences provides access to evolutionary markers that allows more robust evolutionary inferences than viral fragments [32,33]. Herein, we report a genomic investigation and phylogenetic study based on the complete MAYV and CHIKV genomes sequenced from patients living on the southern border of the Brazilian Amazon rainforest.

## 2. Materials and Methods

### 2.1. Study Site

This study was carried out in Sinop city (11°50′53′′ S and 55°38′57′′ W), located 503 km north of the Mato Grosso state capital of Cuiabá, Midwest Brazil. The city was established in 1974, inside the Amazon rainforest, and today, it has an estimated population of 139,935 inhabitants. Sinop is situated in a transitional zone between the Amazon rainforest and Cerrado (Savannah) biomes. The region has a tropical climate, with a dry season of six months, and an average temperature of 26 °C and precipitation levels of over 2500 mm.

### 2.2. Sample Collection 

Serum samples were collected during arboviral surveillance from February 2014 to October 2018. As inclusion criteria, we sought patients presenting clinical diagnoses of dengue, Zika or chikungunya infection, who reported arbovirus-like symptoms, such as fever, myalgia, headache and arthralgia. All of them were interviewed and signed an informed consent form.

### 2.3. Viral Detection—RT-PCR Assays

Viral RNA was extracted directly from the samples or cells’ supernatant using a QIamp viral RNA mini kit (QIAGEN, Hilden, Germany), followed by a reverse transcription using 1 μL of random hexamers (Promega, Madison, MI, USA), 1 μL of Go Script Reverse Transcriptase (Promega) and 8 μL of RNA, for a 20 μL reaction, according to the manufacturers’ instructions. A duplex-PCR was made using the genus-specific primers for alphaviruses (targeting nsP1 region) and flaviviruses (targeting NS5 region), followed by multiplex-semi-nested RT-PCR assays for species-specific identification [34], with a few modifications, as previously described [31].

### 2.4. Viral Detection—Isolation

The MAYV and CHIKV positive samples were inoculated into the cell culture of *Ae. albopictus* C6/36 and Vero cell lines; three passages were performed and then the supernatants were tested for viral nucleic acids by RT-PCR assays following [35].

### 2.5. Viral Genome Sequencing

Two PCR positive samples (one for CHIKV and one for MAYV) were assessed by real-time RT-PCR (rRT-PCR), employing the established protocols for CHIKV [36] and adding the MAYV primers used in the conventional PCR, in order to obtain Ct values to guide the genome sequencing strategy. Total RNA was extracted directly from the serum of the CHIKV positive sample and from the supernatant of the MAYV first passage in C6/36 cells, in order to perform the single-strand cDNA synthesis used for viral genomic amplification. RT-PCR reactions were carried out in a volume of 20 μL using Random Hexamers (Promega, Madison, WI, USA) and ProtoScript II Reverse Transcriptase (New England Biolabs, Ipswich, MA, USA), following the manufacturers’ instructions.

Single strand cDNA from the CHIKV positive serum sample was submitted to whole genome amplification using multiplex-PCR [37], while single-strand cDNA from the MAYV sample was submitted to whole genome amplification by using multiplex RT-PCR, with primers designed using Primal Scheme [38]. Primer design was performed with two complete genomes of MAYV belonging to the L and N genotypes (GenBank accession number KY618127 and KP842812, respectively), both detected in South America (primers can be found in Appendix A). Sequencing libraries were prepared with the Nextera XT Library Prep Kit (Illumina, San Diego, CA, USA), using 2 ng of input cDNA derived from the CHIKV and MAYV multiplex PCRs, following the manufacturer’s instructions. A MiSeq Reagent Kit V3 of 150 cycles (Illumina, San Diego, CA, USA) was used, employing a paired-end strategy, resulting in 75 bp reads separated by 350 bp. Samples were sequenced on an Illumina MiSeq platform at the Technological Platform Core at the Aggeu Magalhães Institute (IAM).

Trimmomatic v. 0.36 [39] was employed to remove low-quality reads, adapter sequences and primers. FastQC was used to assess the quality of the Illumina raw reads. These were subsequently mapped against the MAYV BeH473130/1988 (KY618133) and CHIKV BHI3734/H804698 (KP164568) reference genomes, using Bowtie 2 with the default parameters [40]. Consensus sequences of viral genomes were obtained through the Integrated Genome Viewer software [41] and the sequences were deposited in GenBank with the accession numbers: MH513597 and MT349960. The raw sequenced reads may be found in the European Nucleotide Archive under the project number PRJEB38124.

Single Nucleotide Polymorphisms (SNPs) calls were performed with samtools mpileup and vcf-annotate tools [42] with the following parameters: (vcf-annotate-filter Qual = 20/MinDP = 200/SnpGap = 20), including manual SNP checking.

### 2.6. Phylogenetic Analysis 

Multiple sequence alignment was performed using MAFFT version 7 [43], through the MAFFT online server (http://mafft.cbrc.jp/alignment/software/); maximum-likelihood analysis (ML) was performed using PhyML version 3.0 [44] through the web site at http://www.atgc-montpellier.fr/phyml/. Coding regions corresponding to the complete genomes from Sinop were aligned with all published and available near-complete: CHIKV genomes (>8000 nucleotides), belonging to the WA, ECSA and Asian/Caribbean genotypes, totalizing 930 genomes; and MAYV genomes (>8000 nucleotides), belonging to the L, D and N genotypes, totalizing 66 genomes. We also performed phylogenetic analysis using only Brazilian genomes belonging to the CHIKV ECSA genotype, totalizing 79 genomes. All genomes and data were collected from ViPR [45] (http://www.viprbrc.org/), on 15 April 2020 (accession numbers of all genomes used in the analysis may be seen in Appendix A). The ML phylogenies were reconstructed by using the best-fit general time-reversible (GTR) model with invariant sites (+I) (GTR + I) for MAYV, and 4 gamma substitution rate categories (+G) (GTR + G + I) for CHIKV, all suggested as the most likely models to represent the data by the Smart Model Selection implemented in version 3.0 of PhyML online [46]. For the tree search operation, we used SPR, and statistical support for the phylogenetic nodes was evaluated by an SH-like approximate likelihood ratio test. Tree visualization and figure generation were performed with FigTree v1.4.4 [47].

### 2.7. Ethical Approval

This study was approved by the Research Ethics Committee of the Júlio Müller Hospital-Universidade Federal de Mato Grosso (UFMT) (approval number 288.172/2013) and the Research Ethics Committee of the UFMT (approval number 2.063.295/2018).

## 3. Results

Serum samples were collected from 354 patients exhibiting dengue-like illnesses, within 20 days of the onset of symptoms. Thirty-four patients were positive for alphaviruses—33 tested positive for MAYV (33/354; 9.3%) and one for CHIKV (1/354; 0.3%) (Figure 1). Seventy-eight samples were positive for flaviviruses and are described elsewhere (Appendix A).

The 33 Mayaro fever cases had a median age of 30 years; 18 (54.5%) were male and 15 (45.5%) were female. Clinical characteristics included mainly fever (n = 27), myalgia (n = 20), headache (n = 14) and arthralgia (n = 12). From the MAYV-positive samples, one was sequenced (BR/Sinop/H307/2015), which belonged to a 21-year-old female patient presenting with fever and myalgia, collected on the third day of symptoms, in March 2015. The other sample (BR/Sinop/H542/2018) was from a 20-year-old patient presenting with fever, myalgia, headache, arthralgia, retro-orbital pain and hematemesis, collected on the fourth day of symptoms, in February 2018 (Appendix A). Both patients were urban residents and reported no recent history of travel or access to urban parks, sylvatic or rural areas in temporal proximity to the emergence of the first symptoms.

The cycle thresholds (CTs) for the positive samples were 6.12 and 23.19 for the MAYV and CHIKV samples, respectively. We obtained 2,813,153 reads for the MAYV sample, in which 84.48% mapped to the reference genome, reaching an average coverage depth of 15,453.99; from the CHIKV positive sample, we obtained 1,123,424 reads, in which 96.47% mapped to the reference genome, reaching an average coverage depth of 7133.15 (Table 1). 

The maximum likelihood phylogenetic reconstruction of the obtained MAYV genome and a few other available MAYV genomes showed that the Sinop strain clustered within the L genotype and was closely related to the MAYV strains detected from mosquitoes in the 1960s in Pará state (PA) in the northern region of Brazil (Figure 2a,b), showing a high aLRT SH-like support of 1.0. Meanwhile, the CHIKV genome was placed in a polytomy within a previously described Brazilian subclade that is an offshoot of the East/Central/South African lineage [21], which also had a high aLRT SH-like support of 1.0 (Figure 3a). Interestingly, the CHIKV sequenced here did not cluster with other lineages from the state of Mato Grosso (Figure 3b).

We found a small number of SNPs per sample, varying from 17 in the CHIKV to 30 in the MAYV genome (Figure 3). For CHIKV, six SNPs found were non-synonymous and 11 were synonymous. For MAYV, seven SNPs found were non-synonymous and 22 were synonymous. Moreover, we found several specific mutations restricted to the genomes sequenced in this study: CHIKV presented four new amino acid mutations, two in nonstructural proteins (nsP2-T31I and nsP3-A388V) and two in an envelope protein (E3 T20I and H57R); MAYV presented two, both in the E1 protein (I425S and V427A) (Appendix A; Figure 4).

We detected two previously reported CHIKV amino acid mutations in structural proteins: E2-G60D, which contributes to the increased CHIKV fitness in *Ae. albopictus* and *Ae. aegypti* [48], and E1-T98A, which may increase the CHIKV infectivity for *Ae. albopictus* in the presence of E1-A226V substitution [49]. Five amino acid mutations, that were previously described to be under positive selection and to delineate the genotype L strains on vector infection, were revealed of MAYV. We also found nsP1-L518A, nsP3-A298P, nsP3-V386T, nsP4-A249K and E1-L300T [50].

## 4. Discussion

Arboviruses are widespread and diverse in Brazil and impose a large public health burden on the human population due to the abundance of sylvatic and urban vector species associated with poor sanitation and living conditions. As expected in such tropical environments, the simultaneous circulation and human co-infection of these viruses is common [51], imposing a great socioeconomic impact. Most people recover from arbovirus infection with no major sequelae, although life-threatening conditions can be identified such as hemorrhagic fever, Guillain–Barre syndrome and encephalitis [52]. Among a dozen arboviruses that currently circulate in Brazil and infect humans, dengue virus (DENV) has largely predominated, causing several thousand infections every year in the country. In 2019 alone, around 1.5 million cases were reported, and at least 782 people died from DENV infection [53]. Besides DENV, several other arboviruses have been causing human infection over the years, such as MAYV [8,9], Zika virus (ZIKV), yellow fever virus (YFV) and CHIKV, causing outbreaks with hundreds of thousands of people infected annually [54].

Most of the arboviral diagnoses in developing countries are based on clinical and epidemiological criteria, which leads to biased estimates of infection and produces underreports of less prevalent arboviruses that are not investigated in the molecular diagnostic routine. Such sub-notification can be clearly observed in Sinop from 2015 to 2019, where 7341 cases of DENV, 1324 of ZIKV and 15 of CHIKV were confirmed by laboratory tests, whereas the majority of the negative samples, around 44% of the cases investigated, were not tested for other arboviruses [55] (Appendix A). The confirmation of chikungunya diagnosis by Mato Grosso State Department of Health has been based on MAC-ELISA, while MAYV infection has not been investigated routinely yet. Meanwhile, molecular studies using human samples in Sinop have described the silent circulation of SLEV in 2011 and 2015 [56] and a MAYV annual incidence since 2011 [9]. Adding up to this scenario of simultaneous arbovirus circulation, here, we confirm the circulation of CHIKV and MAYV in human samples through molecular testing and genome sequencing.

Genome-wide phylogenetic analysis of MAYV confirmed the circulation of the L genotype in Sinop city, as previously demonstrated [9]. The obtained genome clustered with a genome from an isolate from the state of Pará (PA). Genotype D (widely dispersed) currently covers a broad geographic area from Trinidad and Tobago to Brazil, Peru, Bolivia, Venezuela, Haiti and French Guiana, with several available full-genome sequences (51 genomes). On the other hand, genotype L (limited) has only a limited number of complete genomes available, consisting of ten isolates obtained over the years 1955 to 1991 from PA and two recent isolates: one from Haiti and another from the State of São Paulo, imported from PA [57]. Due to the limited data, it is not possible to precisely determine the L genotype’s dispersion throughout the Amazonian region and its borders. Other studies have also detected fragments of L and D genotypes in other municipalities in MT [58], whereas molecular studies in other border states with MAYV circulation have not been performed yet [59].

We detected two non-synonymous amino acid substitutions I425S and V427A in the MAYV E1 protein, but whether these substitutions played any role in the cell entry capacity and viral persistence is unknown. Alphavirus glycoproteins E1/E2 mediate host recognition and entry into the cell [60]. These proteins are useful in the development of vaccines and serodiagnostic assays as they hold most of the immunogenic epitopes [61,62]. However, the constant humoral immune pressure creates amino acid variations in these proteins that may lead to viral evasion, influence cross-neutralization activity and allow host-switching [63]. Reverse genetic studies are needed to determine whether any of these substitutions may cause major changes in viral fitness. Continued genome-based molecular investigation of MAYV is recommended in order to understand the spread and maintenance of MAYV in the Southern Amazon region and assess antigenic variations that might impact cross-immunity between alphaviruses and affect the sensitivity of diagnostic tests based on protein binding (immunoassays) [64].

Our CHIKV isolate belonged to the ECSA lineage, sharing 99.8% of its nucleotide identity with isolates from Bahia, a state in Northeast Brazil. The CHIKV genome from Sinop did not cluster with other genomes from MT [65]. The distinct evolutionary divergence from the monophyletic MT clade suggests that CHIKV was introduced into MT by at least two independent CHIKV ECSA variants, which are likely co-circulating in the state [66]. With the current data, the circulation of CHIKV genotypes other than ECSA in MT cannot be ruled out, especially in light of the circulation of the Asian/Caribbean genotype in the Amazon region (states of Pará, Roraima and Amapá), highlighting the need for continuous molecular surveillance in the region.

Genome analysis revealed mutations E1-T98A and E2-G60D in the glycoprotein of the CHIKV strain from Sinop. The E2-G60D amino acid change was reported to contribute to CHIKV’s fitness increase in both *Ae. albopictus* and *Ae. aegypti* [48] and is currently detected in all the Asian/Caribbean and ECSA lineage genomes that are available. Meanwhile, the E1-T98A amino acid change has been found in all available ECSA genomes and may increase CHIKV infectivity for *Ae. albopictus* in the presence of E1-A226V substitution [49]. We did not detect other amino acid changes in the E1 glycoprotein previously associated with CHIKV fitness-enhancing in *Ae. aegypti* (E1-A226V, E1-K211E and E2-V264A) [67] and *Ae. albopictus* (E1-A226V, E2-I211T) [68,69], suggesting that the CHIKV lineage circulating in Sinop likely does not have a higher fitness compared to other CHIKV lineages circulating in Brazil. However, the ECSA genotype is currently spreading all over South America and infecting an abundant and diverse *Ae. aegypti* and *Ae. albopictus* population. Since the E1-T98A amino acid substitution is fixed in all ECSA strains from Brazil, including the one sequenced in this study, if the E1-A226V change occurs, CHIKV’s lineage would likely gain increased fitness in *Ae. albopictus*. In this scenario, transmission of this virus could be exacerbated throughout Brazilian and South American territories, as *Ae. albopictus* is already spread over the entire continent [70]. Finally, other sample specific mutations were also detected in the Sinop CHIKV genome, but further genetic studies are required to understand their importance in the virus’ fitness, if any.

As previously discussed, MAYV is endemic in the Southern Amazon region, where many outbreaks have been reported in the last decade [9,58,71]. Considering that alphavirus antibody cross-immunity has already been reported [72], people living in the Southern Amazon region may have alphavirus cross-immunity due to a previous infection by MAYV. In fact, it has been suggested that infection by CHIKV may confer cross-protection from MAYV, following the use of sera from CHIKV-exposed patients to cross-neutralize MAYV in vitro [73]. Such cross-immunity could be the reason why CHIKV only emerged in Sinop in 2018 and has not caused large outbreaks since then. Likewise, the restricted spread of CHIKV in other MAYV endemic areas of the Amazon has been pointed out [74]. However, a comprehensive serological investigation based on neutralization assays or new diagnostic tools must be performed in order to assess the role of alphavirus cross-immunity in such areas.

It is worth noting that, despite the underreports, the highest CHIKV incidence rate in the central-west region of Brazil has been reported in MT (387.6 cases per 100,000 inhabitants), also in 2018 [23]. Noticeably, reports in Sinop did not follow the same trend, both using PCR diagnostic and serological assays, as most of the cases concentrated in the southern cities of the state (Appendix A). Continuous molecular surveillance is required in order to monitor how the spread of recently emerged viruses will unfold [13].

## 5. Conclusions

Epidemiological surveillance based on genome-scale sequencing of the circulating viral strains is valuable for the prompt detection of adaptive mutations, which is essential for understanding transmission patterns, assessing the risk of emergence and intervening in vector control strategies [75]. In this study, we found new mutations in the strains of MAYV and CHIKV and observed that these strains clustered with genomes from geographically distant Brazilian states, suggesting that their spread occurred through infected patients that traveled between states. Further studies are encouraged in order to follow the spread of these viruses within the Southern Amazon region, in order to further understand the importance of mutations in the maintenance and spread of MAYV and CHIKV. This need is reinforced by the large outbreaks of CHIKV in Brazil and the underreporting of MAYV infections [76,77]. New genomic data can clarify the epidemiological characteristics, such as adaptation to vector spread and impact on human infection, where arboviruses of the same viral family co-circulate and may have cross antibody reactivity.

## Figures and Tables

**Figure 1 tropicalmed-05-00105-f001:**
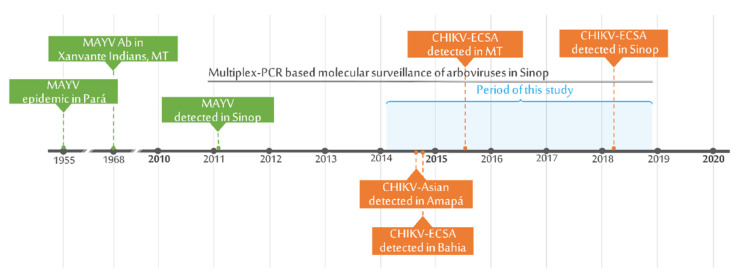
Timepoints of MAYV and CHIKV surveillance/detection in Sinop and nearby areas in the Amazon rainforest and the state of Mato Grosso.

**Figure 2 tropicalmed-05-00105-f002:**
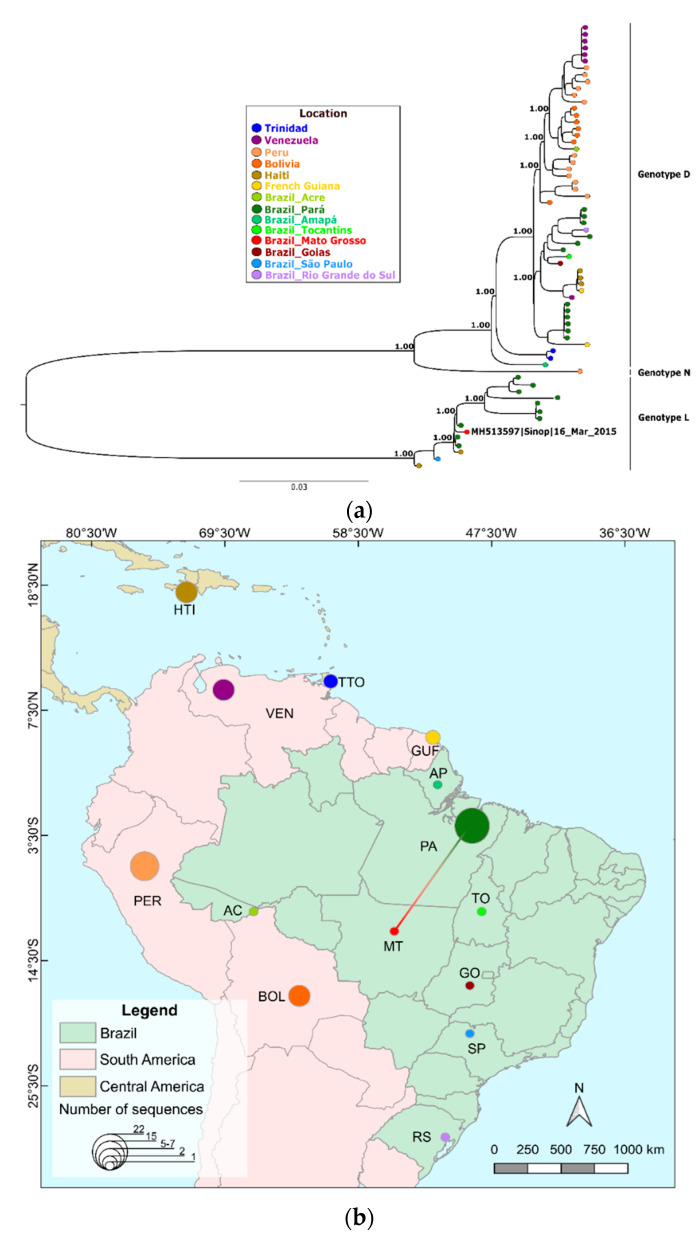
Full phylogenetic reconstruction of MAYV. (**a**) Phylogeny including all available MAYV genomes and positioning of MAYV genome obtained in this study using maximum likelihood of complete and draft genomes (>8 kb) available. (**b**) Distribution of the number of MAYV genomes sequenced per country and Brazilian state and colored edges showing a possible route of MAYV spread based on the most closely related MAYV genome from the phylogenetic analysis shown in panel A. Country and Brazilian state colors follow tip colors in panel A.

**Figure 3 tropicalmed-05-00105-f003:**
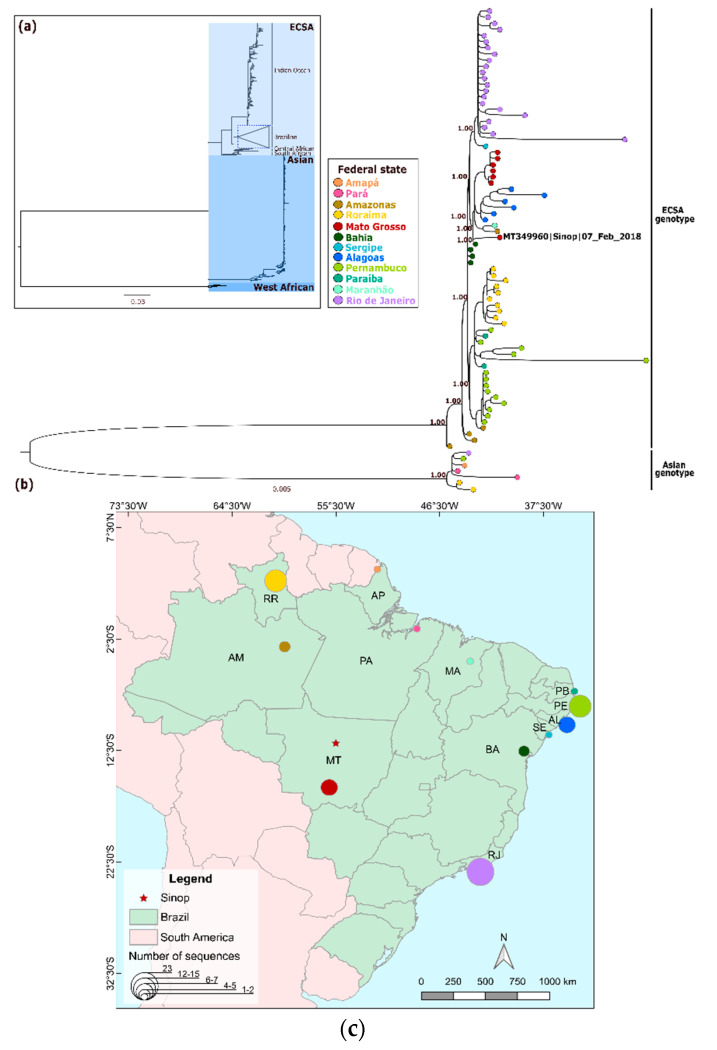
Phylogenetic reconstruction and positioning of CHIKV genome obtained in this study using maximum likelihood. (**a**) Full phylogeny including all available West African, East/Central/South African (ECSA)—Indian Ocean lineages and Asian/Caribbean using complete and draft genomes >8 kb. (**b**) Phylogenetic analysis focusing on Brazilian genomes of a subset of ECSA and Asian/Caribbean CHIKV genomic sequences from the full CHIKV tree. (**c**) Distribution of the number of CHIKV genomes sequenced per state. State colors follow tip colors in panel B.

**Figure 4 tropicalmed-05-00105-f004:**
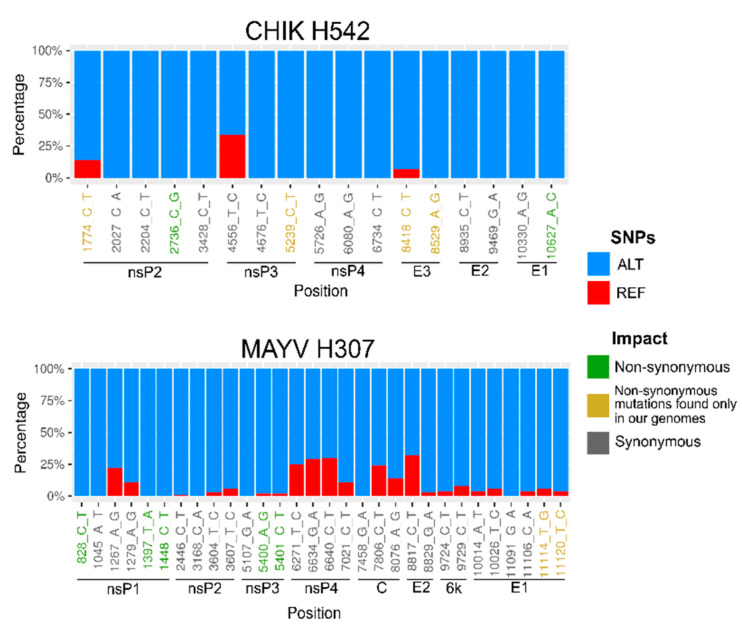
Single nucleotide polymorphism (SNP) analysis. Proportion of sequencing reads supporting the reference nucleotide (REF—red) and the SNPs (ALT—blue) from the CHIKV and MAYV genomes obtained in this study. Green, yellow and gray position names below each bar plot denote non-synonymous, non-synonymous mutations found only in our genomes and synonymous SNPs.

**Table 1 tropicalmed-05-00105-t001:** Epidemiology and sequencing data of two alphaviruses obtained from patients living in Sinop, MT.

Isolate ID	Age	Gender	Onset of Symptoms—dd/mm/yyyy	Collection Date—dd/mm/yyyy	Sample	qRT-PCR Ct ^a^	Genbank Accession Code	Total No. Reads ^b^	No. Mapped Reads (%) ^b^	Depth of Coverage ^b^	Genome Length (nt) ^b^
H307 MAYV	21	F	14/03/2015	16/03/2015	C6/36 passage 1	6.12	MH513597	2,813,153	2,376,618 (84.48%)	15,453.99	11,147
H542 CHIKV	20	F	04/02/2018	07/02/2018	Serum	23.19	MT349960	1,123,424	1,083,817 (96.47%)	7133.15	11,492

Depth of coverage = number of mapped reads × 75 (read length)/reference genome length (11,534 for MAYV and 11,812 for CHIKV). ^a^ Ct, cycle threshold; qRT-PCR, quantitative reverse transcription PCR. ^b^ Genomic sequencing statistics (%) were calculated using KY618133 for MAYV (11,534 nt long) and KP164568 for CHIKV (11,812 nt long) as reference genome.

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
