# Peer review of "The Emergence of Chikungunya ECSA Lineage in a Mayaro Endemic Region on the Southern Border of the Amazon Forest"

_tropicalmed, 2020, doi:10.3390/tropicalmed5020105_

Round 1
Reviewer 1 Report
This manuscript is a well-written, concise, timely and very informative report on the circulation of arthralgic alphaviruses in Sinop city, Brazil. This report investigates the presence of arboviruses in sera of 354 patients sampled from February 2014 to October 2018 in Sinop city. They report on the molecular detection of MAYV and CHIKV, and present the genome sequences and a detailed phylogenetic analysis of both viruses, accompanied by an intuitive discussion on the role of alphavirus cross-reactive immunity on the spread and emergence CHIKV and MAYV. Below are minor comments and suggestions meant to increase the rigor of the manuscript.
Major comments:
Is there any serological data available for CHIKV and MAYV in Sinop city during this sampling period? This can offer support on whether cross-reactive immunity is really playing a role in reducing CHIKV emergence. A very important point to be considered here is that, in the absence of CHIKV seroprevalence data, it may not be appropriate to draw conclusions that MAYV circulation may be limiting CHIKV emergence, when in fact, CHIKV may have already spread widely and therefore there is sufficient herd-immunity in the population so no new cases are being detected.
Do the authors have the capacity to perform serological tests on these sera samples or a subset thereof. This can benefit the work considerably.
The supplementary information is not accessible via the website or the link within the manuscript. This makes it difficult to thoroughly review. To this reviewer, Supplementary table 4 sounds very informative and may be useful in the manuscript. Again, it’s hard to tell if this should be presented here because the complexity is unknown.
Offer statistical support within the text when discussing the separation of the clades and evidence for independent introductions.
Minor comments
Line 24 remove the word virus after CHIKV
Line 45-46 The equine encephalitides are arguable among the most important alphaviruses and should be listed here.
Line 64. Why spell out Chikunguna virus when it was abbreviated previously in the text
Figure 1 legend. 1b needs a scale within the legend to explain the difference in diameter of the spheres and the number of genomes available
Figure 3 can be supplementary if needed.
Author Response
Response to Reviewer 1 Comments
- Is there any serological data available for CHIKV and MAYV in Sinop city during this sampling period? This can offer support on whether cross-reactive immunity is really playing a role in reducing CHIKV emergence. A very important point to be considered here is that, in the absence of CHIKV seroprevalence data, it may not be appropriate to draw conclusions that MAYV circulation may be limiting CHIKV emergence, when in fact, CHIKV may have already spread widely and therefore there is sufficient herd-immunity in the population so no new cases are being detected.
We agree with the reviewer that serological information would be very important to evaluate such a question. Even before CHIKV was introduced in Brazil (2014), our team has been working on the surveillance of arboviruses in the Amazon biome in the northern Mato Grosso, including the use of CHIKV primers on clinical human samples and we did not detect CHIKV until this first sample from 2018. However, we have not performed a serological investigation so far. The only serological data for CHIKV in Sinop city is available from the Mato Grosso State Department of Health, based on the diagnostics of CHIKV suspected cases. This data is shown within the supplementary table 6. While there is no MAYV serological protocol available so far.
We have decided to make this information clearer by stating: "[... ] The confirmation of chikungunya diagnosis by Mato Grosso State Department of Health has been based on MAC-ELISA, while MAYV infection has not been investigated routinely yet.", lines 257-258.
- Do the authors have the capacity to perform serological tests on these sera samples or a subset thereof. This can benefit the work considerably.
Unfortunately we have no current capacity to perform serological tests on these samples now. The surveillance system of the Public Health Services responsible for the application of MAC-ELISA has not attended to the demand either because of the high number of samples accumulated in the official laboratories or the lack of supplies. Our team is currently engaged in the development of serological tests aiming at the specific detection of alphaviruses (CHIKV and MAYV). Since we lack a standardized in-house protocol to perform serological evaluation of our samples, we opted to state in the manuscript, in lines 323-326, that "Likewise, the restricted spread of CHIKV in other MAYV endemic areas of the Amazon has been pointed out [73]. However, a comprehensive serological investigation based on neutralization assays or new diagnostic tools must be performed to assess the role of alphaviruses cross-immunity in such areas and its implication on the eco-epidemiological feature".
- The supplementary information is not accessible via the website or the link within the manuscript. This makes it difficult to thoroughly review. To this reviewer, Supplementary table 4 sounds very informative and may be useful in the manuscript. Again, it’s hard to tell if this should be presented here because the complexity is unknown.
We apologize for failing to provide the supplementary information and for causing inconvenience in reviewing the manuscript. We will upload again the supplementary material and hope that it will be available through the system in the next round of review.
- Offer statistical support within the text when discussing the separation of the clades and evidence for independent introductions.
We agreed with the reviewer and improved the text accordingly.
Minor comments
Line 24 remove the word virus after CHIKV
Changed as suggested.
Line 45-46 The equine encephalitis are arguable among the most important alphaviruses and should be listed here.
We included ‘[…] and the viruses of the Western (WEEV), Eastern (EEEV) and Venezuelan (VEEV) equine encephalitides [13].’ in the lines 46-47.
Line 64. Why spell out Chikungunya virus when it was abbreviated previously in the text
As a matter of style, most journals advise authors to spell out any abbreviation or a number when opening a sentence. Therefore, we decided not to use abbreviations at the beginning of a paragraph or when initiating sentences.
Figure 1 legend. 1b needs a scale within the legend to explain the difference in diameter of the spheres and the number of genomes available.
Scale added as required.
Figure 3 can be supplementary if needed.
As it is not necessary to have it as supplementary, the authors have decided to keep it in the main text.
Reviewer 2 Report
In the manuscript Julia da Silva Pessoa Vieira et al. describe the monitoring for MAYV and CHIKV infections in sera of 345 patients that showed symptoms of arboviral infection. Sampling was from patients in Sinop City, Mato Grosso, Brazil, over the period 2014-2018. 33 MAYV and 1 CHIKV acute infections were detected and 1 clone from each virus was sequenced. The study represents an observational report and some suggestions the authors put forward, such as the introduction of MAYV in the region and the existence of immunity to CHIKV due to cross-reactivity from MAYV immunity are premature given the limited amount of data.
Comments:
- Unfortunately I could not readily access the suppl. data through the journal’s website, and apologize beforehand if remarks would be answered by these.
- Little information about the samples is provided outside of their numbers, did these 345 febrile patients have other detected arboviral infections such as DENV?
- Also, the sampling occurred over a period of 4 years. It would be interesting to know for the reader at which timepoints the MAYV and CHIKV infections were detected in the region. The authors could introduce an extra figure showing a timeline and indicate when the infections took place. It will also allow more informed suggestions about when and from where possible introductions of MAYV or CHIKV occurred in the region.
- The most recent phylogenetic analyses of CHIKV suggest 4 rather than 3 genotypes (Schneider et al., Viruses, 2019: PMID: 31470643) and could be mentioned in the introduction section (line 50)
- In the mat. and meth. Section it is mentioned that single-stranded cDNA from the MAYV sample was submitted to multiplex RTPCR, this probably should read ‘submitted to PCR’
- The mention that arboviral infections cause morbidity and mortality is a dramatic statement, especially since CHIKV and MAYV are commonly associated with little mortality. It would be more measured to quantify those statements with numbers of deaths for example: line 38, line 240
- alphaviral non-structural proteins should be named ‘nsP’ rather than ‘NS’ throughout the text: line 105, line 221, line 234 and in fig. 3
Author Response
Response to Reviewer 2 Comments
- Unfortunately, I could not readily access the suppl. data through the journal’s website, and apologize beforehand if remarks would be answered by these.
We apologize for failing to provide the supplementary information and for causing inconvenience in reviewing the manuscript. We will upload again the supplementary material and hope that it will be available through the system in the next round of review.
- Little information about the samples is provided outside of their numbers, did these 345 febrile patients have other detected arboviral infections such as DENV?
This information is shown in supplementary table 3. Indeed, a total of 78 samples were positive for arboviruses, including DENV (Kubiszeski et al. 2020), SLEV (Moraes et al. 2020), and ZIKV (Vieira et al. 2019), which have been discussed in previous papers. We added such information in the result section as follows “Seventy-eight samples were positive for flaviviruses and are described elsewhere (Supplementary Table 3)”, lines 170-171, and in supplementary table 3.
- Also, the sampling occurred over a period of 4 years. It would be interesting to know for the reader at which timepoints the MAYV and CHIKV infections were detected in the region. The authors could introduce an extra figure showing a timeline and indicate when the infections took place. It will also allow more informed suggestions about when and from where possible introductions of MAYV or CHIKV occurred in the region.
We accepted the suggestion and added an extra figure (Figure 1), line 172.
- The most recent phylogenetic analyses of CHIKV suggest 4 rather than 3 genotypes (Schneider et al., Viruses, 2019: PMID: 31470643) and could be mentioned in the introduction section (line 50)
IOL is suggested to be a genotype in the paper mentioned, but it is known to be a lineage from ECSA genotype, as also supported in our phylogenetic analysis. Therefore, we continue to suggest 3 genotypes rather than 4, as IOL is a lineage derived from ECSA.
- In the mat. and meth. Section it is mentioned that single-stranded cDNA from the MAYV sample was submitted to multiplex RTPCR, this probably should read ‘submitted to PCR’
In fact, we screened all the samples for multiple arboviruses by using multiplex assays (Bonzoni et al., 2005) with modifications (Vieira et al., 2019).
- The mention that arboviral infections cause morbidity and mortality is a dramatic statement, especially since CHIKV and MAYV are commonly associated with little mortality. It would be more measured to quantify those statements with numbers of deaths for example: line 38, line 240
Line 38 changed to: ‘[…] people annually, mainly in South America and Asia continents [3].’
Lines 242-243 changed to: ‘[...] great socioeconomic impact.’
Officially recognized deaths in Brazil are cited in the line 247.
- alphaviral non-structural proteins should be named ‘nsP’ rather than ‘NS’ throughout the text: line 105, line 221, line 234 and in fig. 3
Checked throughout the text and changed to use only ‘nsP’.
Reviewer 3 Report
Manuscript ID: tropicalmed-824713
Type of manuscript: Article
Title: The emergence of chikungunya ECSA lineage in a Mayaro endemic region on the southern border of the Amazon forest
The authors show data on the presence of alphaviruses in sera of 354 patients sampled from February 2014 to October 2018 in Sinop city. They found 33 sequences of MAYV and 1 CHIKV and did phylogenic analysis of the sequences. The main finding was that at least two independent introductions of MAYV occurred in Mato Grosso and that the arrival of CHIKV in Sinop did not cause a surge in human cases in the following years, as observed in the rest of the state, suggesting that cross immunity from MAYV infection might be protecting the population from CHIKV infection.
The data are very interesting and very well presented, although the discussion shows repetition from the results section and should be shortened. Serological data would emphasis the cross-immunity the authors observed and if possible should be added. However, a clear difference between MAYV and CHIKV would require neutralization assays.
Author Response
Response to Reviewer 3 Comments
#Reviewer 3: The data are very interesting and very well presented, although the discussion shows repetition from the results section and should be shortened. Serological data would emphasize the cross-immunity the authors observed and if possible should be added. However, a clear difference between MAYV and CHIKV would require neutralization assays.
We thank the reviewer for such observations. Modifications were made in lines 324-326: “...a comprehensive serological investigation based on neutralization assays or new diagnostic tools must be performed to assess the role of alphaviruses cross-immunity in such areas.”